# Spatial and Temporal Characterization of Drought Events in China Using the Severity-Area-Duration Method

**Xiaoli Yang** [1,2], **Linyan Zhang** [1,2], **Yuqian Wang** [3], **Vijay P. Singh** [4], **Chong-Yu Xu** [5], **Liliang Ren** [1,2,*], **Mengru Zhang** [1,2], **Yi Liu** [1,2], **Shanhu Jiang** [1,2] and **Fei Yuan** [1,2]

[1] State Key Laboratory of Hydrology—Water Resources and Hydraulic Engineering, College of Hydrology and Water Resources, Hohai University, Nanjing 210098, China; yangxl@hhu.edu.cn (X.Y.); zhangly@hhu.edu.cn (L.Z.); zh_mengru@163.com (M.Z.); liuyihhdx@126.com (Y.L.); hik0216@hhu.edu.cn (S.J.); fyuan@hhu.edu.cn (F.Y.)
[2] College of Hydrology and Water Resources, Hohai University, Nanjing 210098, China
[3] Northwest Electric Powerdesign Power Design Institute Co., Ltd. of China Power Engineering Consulting Group, Xi'an 710075, China; hhuzhwyq@163.com
[4] Department of Biological and Agricultural Engineering, Texas A & M University, College Station, TX 77843, USA; vsingh@tamu.edu
[5] Department of Geosciences, University of Oslo, NO-0316 Oslo, Norway; chongyu.xu@geo.uio.no
[*] Correspondence: rll@hhu.edu.cn

**Abstract:** Global climate change not only affects the processes within the water cycle but also leads to the frequent occurrences of local and regional extreme drought events. In China, spatial and temporal characterizations of drought events and their future changing trends are of great importance in water resources planning and management. In this study, we employed self-calibrating Palmer drought severity index (SC-PDSI), cluster algorithm, and severity-area-duration (SAD) methods to identify drought events and analyze the spatial and temporal distributions of various drought characteristics in China using observed data and CMIP5 model outputs. Results showed that during the historical period (1961–2000), the drought event of September 1965 was the most severe, affecting 47.07% of the entire land area of China, and shorter duration drought centers (lasting less than 6 months) were distributed all over the country. In the future (2021–2060), under both representative concentration pathway (RCP) 4.5 and RCP 8.5 scenarios, drought is projected to occur less frequently, but the duration of the most severe drought event is expected to be longer than that in the historical period. Furthermore, drought centers with shorter duration are expected to occur throughout China, but the long-duration drought centers (lasting more than 24 months) are expected to mostly occur in the west of the arid region and in the northeast of the semi-arid region.

**Keywords:** SAD method; SC-PDSI; drought severity; drought center; CMIP5 model; multi-model ensemble

## 1. Introduction

Drought is a natural hazard with a complex mixture of magnitude, duration [1,2], and areal extent of precipitation deficit, and occurs in virtually all climatic regimes. Drought in China is extraordinarily prominent because of various climate types and its unique geographical location that gives rise to tremendous spatial and temporal differences in precipitation [3]. Statistics of the Ministry of Agriculture of China showed that the average area affected by drought in China was approximately 266,666.7 km² per year in 2005–2015. During this period, the losses of grain production equaled 30 billion kilograms, ranking first among the economic impacts of all natural disasters. The

Fifth Assessment Report of the Intergovernmental Panel on Climate Change (IPCC AR5) [4,5] found that the average temperature in China had risen by 0.65–1.06 °C over the last 100 years, and this trend is set to continue, leading to more drought events in the future. Therefore, it is of practical significance to study the temporal and spatial patterns of drought and to predict future drought trends in China to improve forecasting and mitigation strategies [6].

Global climate models (GCMs) have become an essential tool for simulating climate change and have been widely used in simulating large-scale climatic elements. However, single climate models do not accurately simulate changes in temperature and precipitation in a given region due to less-than-accurate calculation methods, discretization of numerical values, and inability to account for heterogeneity within the grid cells [7]. Thus, to improve the accuracy of simulation of future climate change, several studies [8,9] have attempted to overcome the systematic deviation of single climate models by using multi-model ensembles (MMEs) [10], such as simple model averaging (SMA) [11], Bayesian model averaging (BMA) [12,13], weighted ensemble averaging (WEA) [14], and reliability ensemble averaging (REA) [15,16]. Indeed, previous studies [17–20] have found that the simulation performance of multi-model ensembles is generally better than that of single models.

Drought indexes, either based on single or multiple hydrometeorological factors, have been used to identify the causes and severity of drought [21–26]. The standardized precipitation index (SPI) [27], which is based on cumulative precipitation probabilities, has been used to analyze drought in several regions and can improve the detection of onset and closure of individual drought events [28–30]. In addition, De Oliveira-Júnior et al. [31] analyzed drought severity based on the SPI index and its relation to the ENSO and PDO climatic variability modes in the regions North and Northwest of the State of Rio de Janeiro, Brazil.

The Palmer drought severity index (PDSI) [32] is the most widely used index because of its simple data acquisition and calculation requirements. While the PDSI can effectively identify the initiation and termination of drought, some studies have suggested that this approach involves subjective conjectures [33,34]. Wells et al. [35] produced a self-calibrating PDSI (SC-PDSI) to calibrate meteorological data for adopting local climate parameters thus that the SC-PDSI is more comparable across a wider range of contexts.

Nonetheless, we need to pay more attention to the spatial and temporal continuity of drought development (such as spatially contiguous areas under drought) and analysis of their characteristics [36]. Andreadis et al. [37] created a severity-area-duration (SAD) analysis method, which regards drought as an individual event. This method combines severity, area (extent), and duration of drought events to effectively analyze changes in the drought characteristics of different regions (e.g., Xiao et al. [38]; Shao et al. [39]). For example, Sheffield et al. [40] used cluster analysis and the SAD method to analyze global and continental drought characteristics. Wang et al. [41] used the SAD method, based on soil moisture, to identify drought events between 1950 and 2006 in China. Zhai et al. [42] applied the intensity-area-duration method to analyze droughts in China between 1960 and 2013. Liu et al. [43] used the SAD method to analyze the spatial and temporal evolution of drought events between 2000 and 2008 in the Colorado River basin, USA. Shao et al. [39] used SC-PDSI index and SAD method to analyze the drought characteristics of recent decades (1980–2015), and found that both methods can capture well the historical drought events of China.

However, these former studies do not provide insight into the projected behavior of individual drought events across China. Therefore, in this study, we identified the projected extreme drought events in China under RCP4.5and RCP8.5 scenarios, based on SC-PDSI, a cluster analysis method, and the severity-area-duration method. Furthermore, the potential spatiotemporal distribution of those extreme drought centers in the whole of China was estimated by the SAD envelop curve.

## 2. Data and Methods

### 2.1. Datasets

In this study, observational datasets of monthly precipitation and temperature with 0.5° spatial resolution were obtained from the Chinese meteorological data sharing service (http://data.cma.cn/)

(Figure 1). The historical period of CMIP5 models was 1961–2005, and the future projected year was beginning from 2006. For analyzing the decadal characteristics of drought events, we selected models and observed data for the period 1961–2000. We divided the historical period into 4 decades (the 1960s, 1970s, 1980s, 1990s) in order to analyze comparatively the historical SAD envelope curves and drought centers. Various climate types and their unique geographical locations gave rise to tremendous spatial and temporal differences in drought characteristics. China spans many degrees of latitude and has complicated terrain, and, therefore, the climate varies sharply. For investigating the regional characteristics of drought across China, we followed the method of Fu et al. [44] to divide China into 4 climatic regions (arid, semi-arid, semi-humid and humid) based on the temporal and spatial distribution of precipitation in China (Figure 1), to analyze the characteristics of drought events over China.

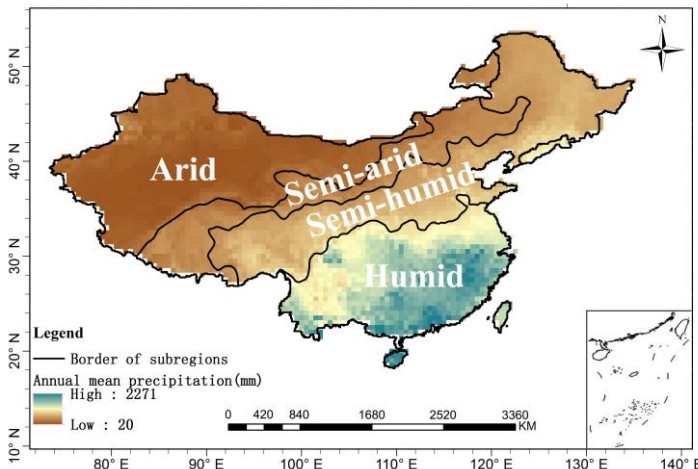

**Figure 1.** Four climate categorizations and annual mean precipitation (mm) in China produced using gridded data with 0.5° spatial resolution based on more than 2400 national meteorology stations.

Limited by data availability, GCMs cannot represent well the characteristics of topography land surface over China, 9 CMIP5 (https://esgf-node.llnl.gov/projects/cmip5/) models were selected to project future monthly precipitation and temperature under the moderate emission path scenario (RCP4.5) and high emission scenario (RCP8.5) (Table 1). To improve the spatial resolution of different GCMs, data from the 9 climate models were downscaled and bias-corrected to a 0.5° × 0.5° resolution using the equidistant cumulative distribution function (EDCDF) quantile-based mapping method [45,46].

**Table 1.** Details of the nine climate models used in this study.

| NO | Model Name | Country | Institute | Resolution (lat × lon) |
|----|-----------|---------|-----------|------------------------|
| 1 | ACCESS1-0 | Australia | Commonwealth Scientific and Industrial Research Organisation (CSIRO) and Bureau of Meteorology (BOM) | 1.25° × 1.875° |
| 2 | CCSM4 | Canada | National Center for Atmospheric Research | 0.9° × 1.25° |
| 3 | CNRM-CM5 | France | Centre National de Recherches Meteorologiques and Centre Europeen de Recherche et Formation Avancees en Calcul Scientifique | 1.4° × 1.4° |
| 4 | HadGEM2-AO | United Kingdom /South Korea | Met Office Hadley Centre/National Institute for Medical Research | 1.25° × 1.875° |
| 5 | MIROC5 | Japan | Atmosphere and Ocean Research Institute (The University of Tokyo), National Institute for Environmental Studies, and Japan Agency for Marine-Earth Science and Technology | 1.4° × 1.4° |
| 6 | MPI-ESM-LR | Germany | Max Planck Institute for Meteorology | 1.865° × 1.875° |
| 7 | MRI-CGCM3 | Japan | Meteorological Research Institute | 1.12° × 1.125° |
| 8 | NorESM1-M | Norway | Norwegian Climate Centre | 1.9° × 2.5° |
| 9 | NorESM1-ME | Norway | Norwegian Climate Centre | 1.9° × 2.5° |

## 2.2. Modeling Methods

### 2.2.1. Bias-Correction and Multi-Model Ensemble Method

The EDCDF method [45] was used to bias-correct the CMIP5 model outputs over China to improve spatial resolution and accuracy. This method constructed the cumulative distribution function (CDF) of the simulated historical values and the future simulated values of different climate elements, respectively. It is efficient at improving the inherent errors of climatic model data. It improves on previous approaches based only on the historical CDF because it takes into account any changes in the future distribution [47]. Equation (1) is used to bias-correct the future GCMs simulations of the temperature, adopting the beta distribution with 4 parameters (Equation (2)):

$$x_{m-p_{\_adjust}} = x_{m-p} + F_{o-t}^{-1}\left(F_{m-p}(x_{m-p})\right) - F_{m-t}^{-1}\left(F_{m-p}(x_{m-p})\right) \tag{1}$$

$$f(x; a, b, p, q) = \frac{1}{B(p,q)(b-a)^{p+q-1}}(x-a)^{p-1} \cdot (b-x)^{q-1}, \ a \leq x \leq b; p, q > 0 \tag{2}$$

where B is the beta function, a and b are the range parameters as the extreme values from the data, extended by a certain percentage of the standard deviation, and p and q are the shape parameters determined by the maximum likelihood estimation method.

Equation (3) is used to bias-correct precipitation, with a 2-parameter mixed gamma distribution (Equation (4)) considering the intermittent nature of precipitation:

$$x_{m-p_{\_adjust}} = x_{m-p} \frac{F_{o-t}^{-1}\left(F_{m-p}(x_{m-p})\right)}{F_{m-t}^{-1}\left(F_{m-p}(x_{m-p})\right)} \tag{3}$$

$$f(x; k, \theta) = x^{k-1} \frac{e^{-x/\theta}}{\theta^k \Gamma(k)} \qquad for \ x > 0 \ and \ k, \theta > 0 \tag{4}$$

where $x_{m-p}$ is the model projection value; $x_{m-p_{\_adjust}}$ is the adjusted model projection value after bias-correction; $F_{o-t}^{-1}$ and $F_{m-t}^{-1}$ are the quantile functions corresponding, respectively, to the observations (*o*) and simulations (*m*) in the training period (*t*); and $F_{m-p}$ is the CDF of model simulated fields. Further details about this method can be found in Yang et al. [46] and Li et al. [47]. In the EDCDF method, the parametric distributions are fitted to both temperature and precipitation fields for each grid point.

To better improve the reliability of future projections from GCMs' outputs, multi-model ensemble (MME) methods have been proposed that distilled the uncertainty across models in simulating the climate [9]. The Simple Model Averaging (SMA) method was the simplest and widely used multi-model ensemble technique [48]. Therefore, we applied the SMA method to the 9 CMIP5 models to form the 9-model ensemble (9ME).

### 2.2.2. SC-PDSI

The self-calibrating PDSI (SC-PDSI) [35] can be used to adopt local climatic parameters for different stations, making the PDSI spatially comparable across a larger area. The calculation of the SC-PDSI includes hydrological revenue, expenditure, and standardization. At the heart of the SC-PDSI is the water balance equation under a suitable climate:

$$\tilde{P} = \alpha_i PET + \beta_i PR + \gamma_i PRO - \delta_i PL \tag{5}$$

$$\alpha_i = \frac{\overline{ET_i}}{\overline{PET_i}}, \quad \beta_i = \frac{\overline{R_i}}{\overline{PR_i}}, \quad \gamma_i = \frac{\overline{RO_i}}{\overline{PRO_i}}, \quad \delta_i = \frac{\overline{L_i}}{\overline{PL_i}} \tag{6}$$

$$d = P - \tilde{P} \tag{7}$$

where $\tilde{P}$ is the number of precipitation events required to maintain a normal soil moisture level for a month; the values of 8 determinant weighting factors were evapotranspiration (*ET*), recharge (*R*), runoff (*RO*), water loss (*L*), potential evapotranspiration (*PET*), potential recharge (*PR*), potential runoff (*PRO*), and potential loss (*PL*); $\alpha$, $\beta$, $\gamma$, and $\delta$ were the water-balance coefficients, which were used to achieve potential values that are climatically appropriate for existing conditions (*CAFEC*); and *d* is the moisture departure, which is the difference between actual precipitation and the computed *CAFEC* precipitation.

Different values of *d* were ascribed to different times and locations, which prevent comparisons between them, used *K*, a climatic characteristic, to correct the moisture departure as follows:

$$K' = 1.5 log_{10} \left( \frac{\frac{\overline{PET_i} + \overline{R_i} + \overline{RO_i}}{\overline{P_i} + \overline{L_i}} + 2.8}{\overline{D_i}} \right) + 0.5 \tag{8}$$

$$K = \begin{cases} K'\left(\frac{-4.00}{2^{nd}} percentile\right), & d < 0 \\ K'\left(\frac{4}{98^{th}} percentile\right), & d \geq 0 \end{cases} \tag{9}$$

where *K'* is the PDSI approximation of the climate characteristic of a region; $\overline{PET_i}, \overline{R_i}, \overline{RO_i}, \overline{D_i}$ are the average potential evapotranspiration, recharge, runoff, and moisture departure; the 2nd percentile represents the possibility of extreme drought that corresponds to a PDSI of −4.00; the 98th percentile is defined as the possibility of extreme waterlogging in the case of a PDSI of 4.00.

The moisture anomaly index *Z* is determined using *d* and *K*, as follows:

$$Z = K \times d \tag{10}$$

### 2.2.3. Identification of Drought Using Cluster Algorithm and the SAD Method

The clustering algorithm was used to identify spatial and temporal variations of drought [37]. In this study, the cluster algorithm combined spatially and temporally contiguous regions with SC-PDSI below the specified (−2) value. It was noted that a drought event could break into several smaller droughts and that multiple droughts could merge into a larger drought for a specified minimum area (25,000 km²). Sheffield et al. [40] did an initial experiment with 25,000 km² and 100,000 km² area thresholds and found that droughts could shrink to a few cells and last multiple years through tenuous spatial connectivity. In order to avoid this situation, within the 150,000 km² area, approximately 60 cells were selected as the minimum area threshold of drought events, and a drought

event with an area of less than 150,000 km² was ignored. This threshold value was the same as that selected by Wang et al. [41] for China.

Drought events occurred across a continuous area with a certain duration. The SAD method developed by Andreadis et al. [37] combined severity, area (extent), and duration of droughts to assess the characteristics of events with different durations and their development in time and space. The SAD method, which is based on the spatial proximity of grids, applies a simple clustering algorithm to identify drought events. Based on SC-PDSI, the severity (S) was calculated as:

$$S = 1 - \frac{\sum SC - PDSI_n}{t} \tag{11}$$

where *t* is drought duration. *SC-PDSI$_n$* is the standardized *SC-PDSI* (*SC-PDSI$_n$*$=10^{\frac{SC-PDSI}{2}}$).

In identifying drought events, severity was first calculated for each grid cell, and the grid cell with the maximum severity to be the center of the drought. The presence or absence of drought conditions in the neighboring grid cells of the drought center was then assessed. If drought conditions applied to a neighboring cell (i.e., where SC-PDSI < −2), these two cells were regarded as the new drought center. The average severity of these two cells was then applied for the new drought center, and the area of two grids was also averaged. This process was then repeated until no drought conditions were detected in neighboring cells. This procedure was used to identify all other drought events occurring in the study period.

When identifying drought events, the average drought severity of an event with different intervals (3, 6, 9, 12, 24, and 48 months) and areas can be determined. The SAD envelope curves of all drought events were also be formed by choosing the maximum severities of all events for each area increment, which reflected the largest drought severity in different drought-affected areas.

## 3. Results

### 3.1. Performance of Climate Models

We used the Taylor diagram to evaluate the performance of the 9 individual models and 9ME (Figure 2). Taylor diagrams make full use of the transformation relationships of the correlation coefficient (CC), normalized standard deviation (NSD), and root-mean-square error (RMSE). The closer CC and NSD are to 1, the more similar the simulation and observation data are. The smaller the RMSE, the smaller the deviation between simulation and observation data. Figure 2 shows that the CC values of the uncorrected models were similar to the bias-corrected models, but the NSD and RMSE performances of the uncorrected models were worse than that of the bias-corrected models. It indicated that overall the corrected models performed better. For further evaluating the performance of 9ME, the spatial distribution of simulated bias was shown in Figure 3. The uncorrected 9ME data had extremely large biases of more than 150 mm/month and 10 °C for precipitation and temperature, respectively, in northwest China (Figure 3a,b). In contrast, the bias-corrected 9ME simulation data (Figure 3c,d) contained biases mostly in the range of −0.8 to 0.8 mm/month. The largest bias (up to 0.8 mm/month) was located in the southwest part of the Qinghai–Tibet Plateau, for which the temperature bias was less than 0.1 °C.

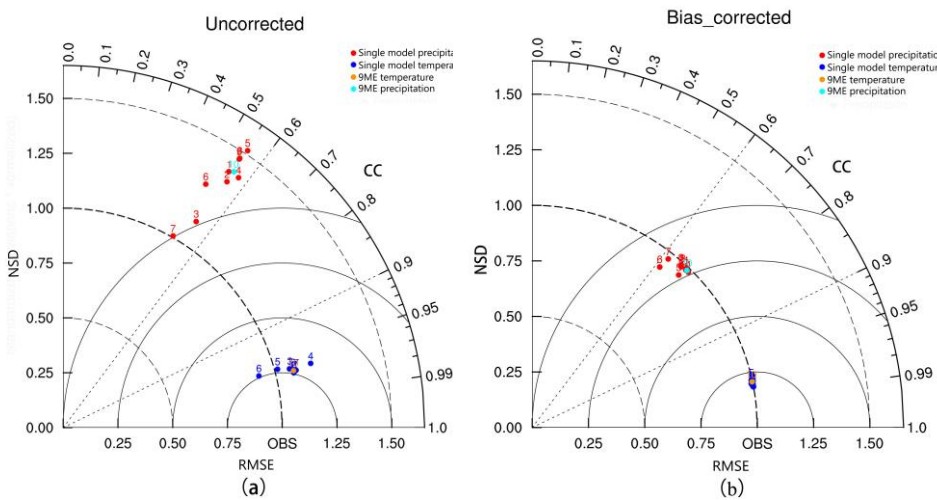

**Figure 2.** Taylor diagrams of uncorrected (**a**) and bias-corrected (**b**) precipitation and temperature of 9 CMIP5 models and 9ME.

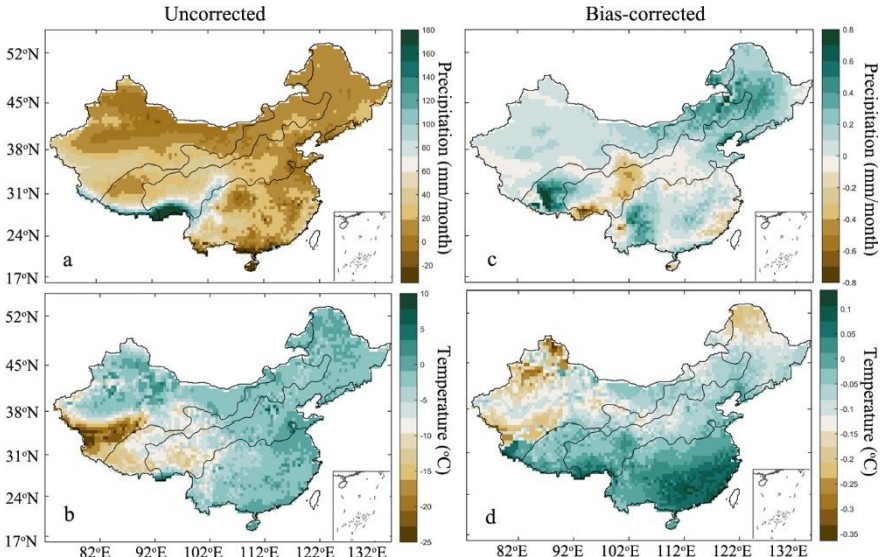

**Figure 3.** The distribution of bias between observed and uncorrected 9ME (**a**,**b**) and bias-corrected 9ME (**c**,**d**) monthly mean precipitation (mm/month, upper panel), and temperature (°C, lower panel) in China.

It should be noted that the accuracy of the bias-corrected single model and the 9ME was higher than that of the uncorrected models. This demonstrated that the bias-corrected 9ME had a good capability in simulating precipitation and temperature over China and that it effectively captured the spatial characteristics of climatological elements.

*3.2. Spatial and Temporal Characteristics of Drought Events*

3.2.1. Historical Period

Based on the SAD method, 49 drought events were identified in China between 1961 and 2000, 17 of which lasted more than 12 months, and 4 of which lasted more than 48 months. Table 2 presents the 6 most severe drought events according to ranked duration, area extent, severity, and comprehensive index. The comprehensive index (CI) was calculated as the area affected by drought multiplied by the average severity of the drought-affected area for each month. The most severe

drought occurred in April 1963, with a severity value of more than 0.9. The longest duration drought was 121 months, lasting from 1964 to 1974, closely followed by the 1982 to 1991 drought (lasting 112 months). The drought of September 1965 and September 1986 occurred across a large area (approximately 45% of China), while the area affected by other droughts was lower (approximately 25% of China). Two of the major drought events we identified (1974–1979 and 1961–1964) agreed with previous assessments [41].

**Table 2.** Six most severe drought events in terms of duration, spatial extent, average severity, and comprehensive index from 1961 to 2000. In the second column, the maximum monthly area of drought and corresponding dates are given in parentheses. In the third column, the monthly maximum severity and corresponding date are given in parentheses. In the fourth column, the maximum comprehensive index and corresponding date are given in parentheses. Top four prominent drought events in terms of the ranked comprehensive index and durations (Bold type) were selected for further analysis.

| Durations (months) | Spatial extent (%) | Severity (0–1) | CI (Area × Average severity) |
|---|---|---|---|
| 1964–1974 (121) | 1964–1974 (47.07%, 09/1965) | 1962–1963(0.92, 04/1963) | **1964–1974 (29.97, 06/1966)** |
| 1982–1991 (112) | 1982–1991 (42.40%, 09/1986) | 1963–1964 (0.89, 05/1963) | **1982–1991 (28.58, 08/1986)** |
| 1976–1985 (103) | 1961–1964 (28.77%, 06/1963) | 1980–1980 (0.86, 03/1980) | 1961-1964 (20.13, 06/1963) |
| 1994–1998 (50) | 1994–1998 (25.33%, 09/1994) | 1980–1981 (0.84, 03/1980) | **1994–1998 (13.49, 09/1994)** |
| 1974–1978 (45) | 1998–2000 (24.13%, 09/2000) | 1969–1970 (0.80, 12/1969) | 1991–1993 (13.17, 11/1992) |
| 1978–1981 (37) | 1974–1978 (21.97%, 07/1975) | 1961–1964 (0.78, 04/1963) | **1978–1981 (11.57, 08/1978)** |

Table 2 also summarizes the top four prominent drought events based on the comprehensive index and durations. Figure 4 shows the three-dimensional spatial distribution of drought area and severity of most prominent events for every ten months of the data record. During its early stages, the drought event of 1964–1974 was mainly distributed across northwest China (western and southeastern Xinjiang, Gansu, and Inner Mongolia) before becoming widespread throughout the whole of China. Towards the end of this drought event, the affected area and severity had decreased significantly.

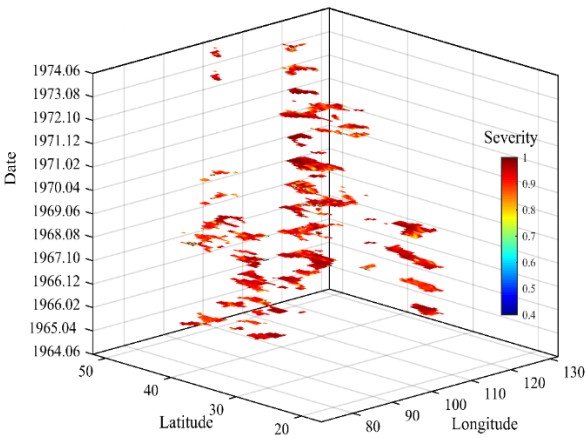

**Figure 4.** Spatial and temporal evolution of the most serious drought events in China between 1961 and 2000.

### 3.2.2. Future Period

For the future period (2021–2060) under RCP4.5, 31 drought events were predicted for China (Table 3), including 9 events with a duration of more than 12 months and 3 with a duration of more than 24 months. The longest duration event of 2021–2042 (253 months) would occur across 68.99% of China. Meanwhile, the most severe drought was projected to occur in February 2026, with a severity

value of more than 0.86. Under the RCP 8.5 scenario (2021–2060), 39 droughts were identified (8 more events than projected under RCP4.5). The longest duration projected drought was 230 months (2021–2041), which was extraordinarily similar to the 2021–2042 events modeled under RCP 4.5. The area of this drought would cover approximately 72.58% of China and would have a severity above 0.9. The modeled duration, area, and severity of all other drought events were significantly reduced relative to this single event.

**Table 3.** Six most severe drought events in terms of duration, spatial extent, average severity, and comprehensive index from 2021 to 2060 under RCP4.5 and RCP8.5 scenarios. The top four prominent drought events in terms of ranked comprehensive index and duration (bold type) were selected for further analysis.

| RCPs | Durations (month) | Spatial extent (%) | Severity (0–1) | CI (Area × Average severity) |
|---|---|---|---|---|
| RCP4.5 | 2021–2042 (253) | 2021–2042 (68.99%, 08/2024) | 2021–2042 (0.86, 02/2026) | **2021–2042 (55.46, 08/2024)** |
| | 2055–2060 (66) | 2055–2060 (20.85%, 07/2057) | 2027–2027 (0.85, 06/2027) | **2048–2049 (10.34, 11/2048)** |
| | 2043–2045 (27) | 2048–2049 (15.64%, 10/2048) | 2022–2023 (0.79, 12/2022) | **2055–2060 (7.80, 08/2057)** |
| | 2050–2052 (21) | 2051–2052 (14.23%, 11/2051) | 2053–2053 (0.79, 06/2053) | **2051–2052 (5.33, 11/2051)** |
| | 2048–2049 (18) | 2055–2056 (12.64%, 07/2055) | 2023–2023 (0.73, 11/2023) | 2054–2055 (5.21, 09/2054) |
| | 2034–2035 (16) | 2054-2055 (8.76%, 09/2054) | 2046–2046 (0.72, 08/2046) | 2046–2047 (5.18, 09/2046) |
| RCP8.5 | 2021–2041 (230) | 2021–2041 (72.58%, 08/2033) | 2021–2041 (0.91, 10/2025) | **2021–2041 (65.57, 10/2025)** |
| | 2051–2056 (61) | 2051–2056 (15.46%, 06/2053) | 2021–2022 (0.91, 01/2022) | **2051–2056 (7.67, 07/2054)** |
| | 2041–2043 (25) | 2056–2057 (10.58%, 04/2057) | 2023–2024 (0.87, 11/2023) | **2041–2043 (7.08, 10/2042)** |
| | 2039–2041 (22) | 2041–2043 (10.48%, 10/2042) | 2024–2024 (0.83, 08/2024) | **2056–2057 (5.44, 04/2057)** |
| | 2045–2046 (18) | 2045–2046 (7.12%, 08/2045) | 2029–2030 (0.79, 01/2030) | 2039–2041 (4.34, 07/2040) |
| | 2056–2057 (15) | 2045–2046 (6.89%, 10/2045) | 2056–2057 (0.78, 09/2057) | 2023–2024 (3.78, 01/2024) |

### 3.3. SAD Envelope Curve of Drought in China

The SAD envelope curve represents the maximum bounds of severity from all drought events of each duration and area increment. For comparative analysis of decadal variation of drought events across China, we divided 1961–2000 into 4 decades, namely the 1960s, 1970s, 1980s, and the 1990s [49]. Figure 5 plots the SAD envelope curve of all drought events with durations of 3, 6, 9, 12, 24, and 48 months within these periods. The severity of drought events weakened with an increase in the spatial extent. Furthermore, the drought events in the 1960s were the most serious on the basis of severity and spatial extent, followed by those occurring in the 1980s.

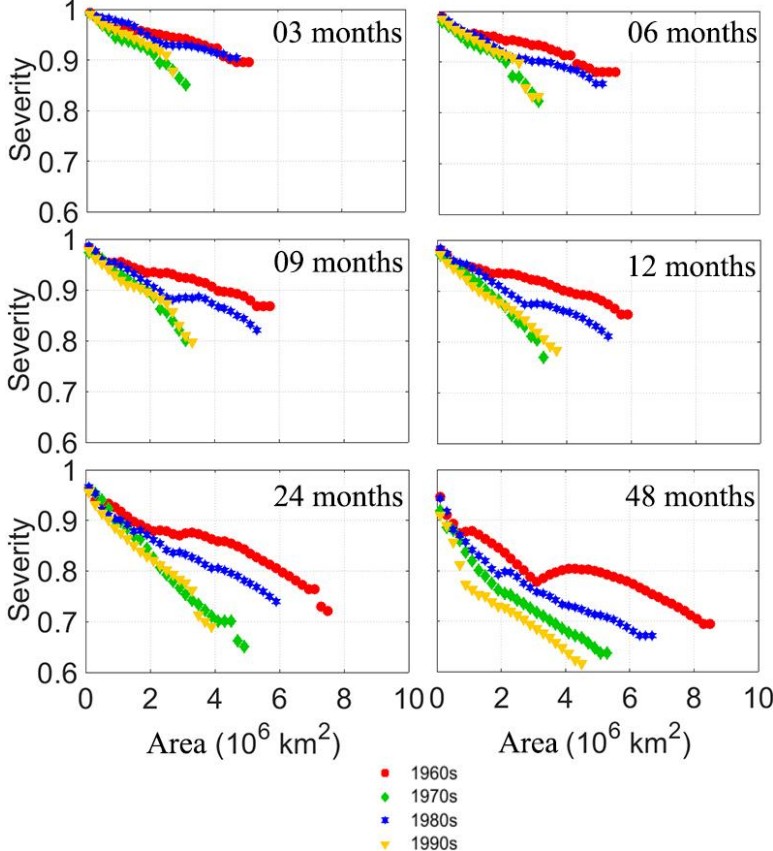

**Figure 5.** Decadal-scale severity-area-duration (SAD) envelope curves for drought events in China with a duration of 3, 6, 9, 12, 24, and 48 months for the period 1961–2000.

It was noted that the drought events in the 1960s dominated the SAD envelope curves for areas up to $5 \times 10^6$ km². This was also observed by Wang et al. [41], who analyzed soil moisture drought in China based on using the SAD method. For drought areas of less than $1 \times 10^6$ km², the most serious events lasting 3 to 12 months occurred in the 1980s. The SAD envelope curves of the 1970s and 1990s events showed relatively small areal extents and lower severity, but the slopes of these envelope curves were significantly steeper than for the other decades.

Figure 6 shows the SAD envelope curves and drought events for the historical period and for the future period based on the two RCPs. The envelope curves under the two RCPs were above the historical envelope curves for all durations, and their spatial extents were also larger than those of the historical period. However, the slopes for the two RCPs were smaller than for the historical period. This indicates that future droughts (under both RCP scenarios) would be more severe than those that occurred during the historical period.

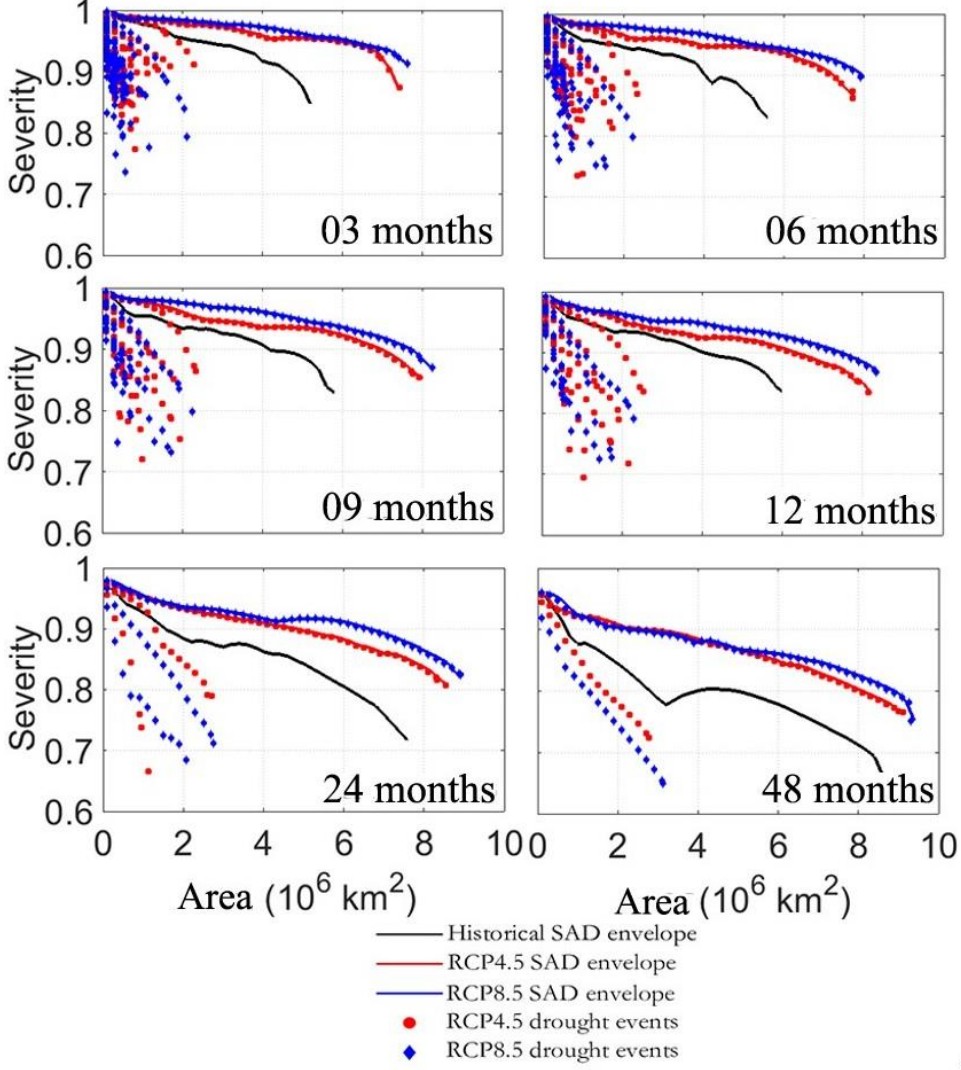

**Figure 6.** SAD envelope curves for droughts with durations of 3, 6, 9, 12, 24, and 48 months during the historical period (1961–2000) and under future RCP4.5 and RCP8.5 scenarios.

### 3.4. Distribution of Drought Centers

The drought centers of all historical drought events (1961–2000) with different durations are shown in Figure 7. We found that during the early decades of the study period, drought centers were mainly distributed in the northwestern and northern regions of China. This was in line with Zhai et al. [42], who analyzed the spatial distribution of drought centers for different durations (1, 3, 6, 9, 12, 24 months) in China for the period 1960–2013. Meanwhile, most drought centers for the late 20th century were located in the semi-arid and southeast parts of the humid regions.

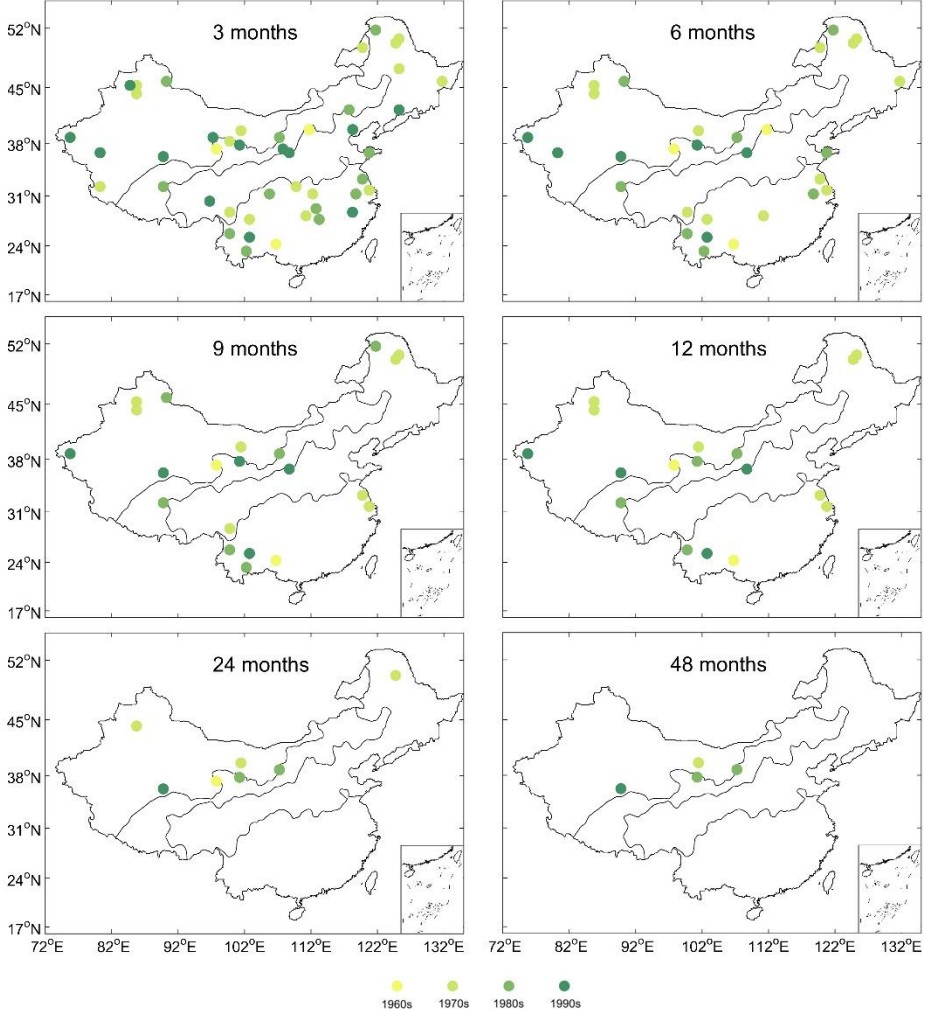

**Figure 7.** Distribution of drought centers with different durations in China during the period 1961–2000.

Furthermore, the short-duration (less than 6 months) drought centers were distributed throughout China. The drought centers for events with 9- and 12-month durations principally occurred during the 1970s and the 1980s, and these were mostly located in the middle of the semi-arid region and the southwest of the arid region. The longer-duration (more than 24 months) drought centers occurred at the boundary of the semi-arid and arid regions, and there were fewer of these compared with short-duration drought centers.

The distributions of drought centers under RCP 4.5 and RCP 8.5 scenarios are shown in Figure 8. The modeled drought centers with short durations (less than 6 months) were distributed throughout China, similar to the observed droughts during the historical period. However, the 9- and 12- month drought centers modeled under two RCPs showed different spatial patterns. Under RCP4.5, drought centers mainly occurred in the northwest of the arid region and in the northeast of the semi-humid region and were more concentrated than under RCP8.5. By comparison, the historical drought centers with the same durations were mostly located at the boundary of the arid and semi-arid regions, and in the southwest of the humid region. This indicates that the frequency of drought events would increase over northeastern China in the future. In this respect, we achieved the same result as Zhai et al. [50] who analyzed the drought trends using the PDSI and the SPI. Drought centers with 24- and 48-month durations were predicted to occur mostly during the 2030s and the 2040s, and the distribution of these drought centers under the two RCPs was roughly similar to the moderate duration events (9- and 12- month), although this was clearly different from the historical period.

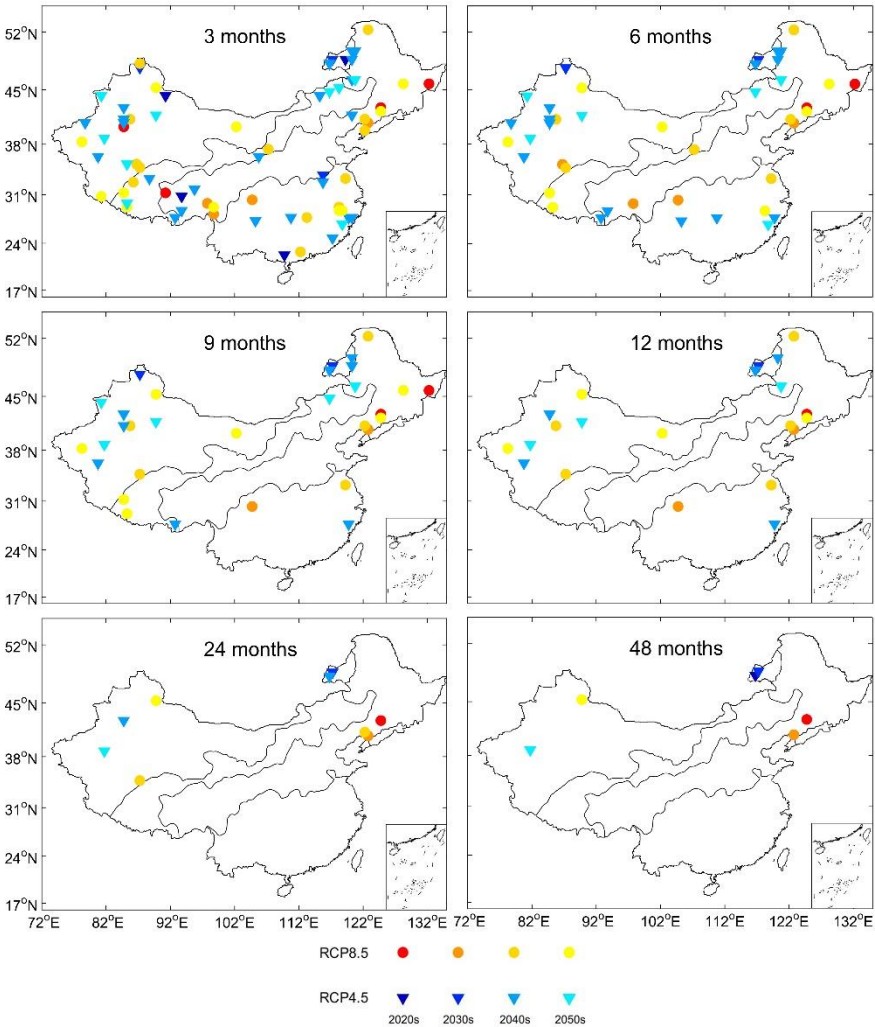

**Figure 8.** Distribution of drought centers with different durations in China under two RCP scenarios for the period 2021–2060.

## 4. Conclusions and Discussion

Using bias-corrected 9ME simulations, we applied the SC-PDSI index and the SAD method to identify drought events and analyze the potential spatiotemporal distribution of those extreme drought centers over China. The primary purpose of this study is to provide a historical perspective when planning for future drought mitigation. We found that historical drought events with short durations occurred throughout China, while the long duration droughts were more frequently located in the boundary region of the semi-arid and arid regions and the southwest of the arid region. The 1970s and the 1990s events showed relatively small areal extents and lower severity than did other decades. It demonstrates that more drought events in the late-20[th]-century affected a smaller area than in the mid-twentieth century. This is similar to Chen et al. [49] that droughts over China exhibited a well-defined decadal variation during the past 50 years, with more frequent droughts occurring before the 1980s and in the 2000s and fewer droughts in the 1980s and 1990s.

Furthermore, our findings indicate an overall increasing risk of droughts over China during the historical period, which hints that climate change and the East Asian monsoon play an important role in drought events. The East Asian monsoon has weakened in terms of land-ocean pressure gradients over the past 30 years [51], and the precipitation rain belts are shifting southward. Therefore, further studies are needed to predict the effects of East Asian monsoon on future drought characteristics in China.

In the future, drought events are projected to occur less frequently than the historical period. The longest modeled drought duration, lasting 253 months during the period of 2021–2042, would affect 68.99% of China under RCP4.5, followed by the drought of 2021–2041 under RCP8.5. It indicates that a drought would occur with a long duration and strong severity in the 2020s–2040s. The projected middle-duration drought centers under two RCPs were mainly located in the northwest of the arid region and the northeast of the semi-humid region of China.

In this study, based on bias-corrected CMIP5 models, we used the SAD method to assess whether future climate change trends are changing (or have the potential to alter) the severity of drought occurrence across China. Our results indicated that the outputs of the bias-corrected multi-model ensemble had high accuracy in simulating precipitation and temperature in China. However, climate models have uncertainty for the theoretical understanding of climate change remains incomplete with certain simplifying assumptions [52,53]. Therefore, future research, particularly regarding drought projections using more climate models and an analysis of narrowing models uncertainties, is essential for a better understanding of future drought characteristics changes. In addition, large-scale teleconnection patterns, such as El Niño–Southern Oscillation (ENSO), have an effect on drought occurrence [54–56], which may also affect the drought prediction over China. For example, Zhang et al. [57] found that more frequent drought struck in southern China during autumn in the two most recent decades and the increasing autumn drought is largely attributed to an ENSO regime shift. Future studies need to pay more attention to the effects of teleconnection and the occurrence of drought. Furthermore, we used 150,000 km$^2$ [41] as the minimum area threshold of drought events for China. Due to the threshold selection that has direct effects on the identification of drought events [40], the later application for other regions needs caution when selecting the threshold.

**Author Contributions:** All authors contributed to the design and development of this manuscript. X.Y. designed the analysis framework and computer programs; L.Z. carried out the data analysis and prepared the first draft; Y.W. download the data and carried out the modeling work; V.P.S., C.Y.X., L.R., M.Z., Y.L., S.J., and F.Y. verified the overall reproducibility of results and other research outputs.

**Funding:** This work was supported by the National Key Research and Development Program under Grant 2016YFA0601504 approved by Ministry of Science and Technology of the People's Republic of China, the National Natural Science Foundation of China (51579066 and 41807165).

**Acknowledgments:** We thank the climate modeling groups (listed in Table 1 in this paper) for producing the models we used and making their model outputs available. We also appreciate the Chinese meteorological data sharing service (http://data.cma.cn/) for providing the precipitation and temperature observed data.

**Conflicts of Interest**: The authors declare no conflict of interest.

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
