# Peer review of "Spatial and Temporal Characterization of Drought Events in China Using the Severity-Area-Duration Method"

_water, doi:10.3390/w12010230_

Round 1
Reviewer 1 Report
The manuscript provides past and future drought scenario over China by using observational (monthly temperature and rainfall) and model data. The manuscript is well organized however there are some questions still need to be clarified and explained by authors:
Page 2, Line 77: Check the sentence for grammatical meaning Page 3, Lines 95-96: Why only 9 models of CMIP5 are used and why not other models. Any specific reason for selecting these 9 models only?! Page 3, Lines 109-112: Any reason for the usage of different methods in bias-correction to temperature and precipitation. Page 5, Line 155: what is the threshold for the highest severity Page 9, Table 3. In table 3 is “PRC4.5” or “RCP4.5”

Reviewer 2 Report
The paper analyses extreme drought events in China using Palmer drought severity index and severity area duration. Different model outputs have been used and the future evaluation of drought events under different scenarios have been defined.
The manuscript take on the problem of drought events and their evolution in space and time. The subject is interesting and current but the paper is not enough further analysed. The description of data, method, and results is superficial and it is not explain entirely methods, selection criteria and data used.
The most lack is about the historical period used, 1961-2000, without explanation because it is not 1971-2010 or 1981-2010 and without explanation because data since 2000 to 2018 are not used. In addition, the prediction of period 2021-2060 has been did without calibration. The period 2001-2018 could be used for calibration model and successively use calibrated model to define future drought events. Without calibration of the model, it is not possible to say if results are correct and adequate or not.
In addition, the paper is not easy to read, some sentences are not understood and there are inscrutable periods.
The subject of the paper is interesting but it is not enough further analysed. The description of data, method, and results is superficial and it is not explain entirely methods, selection criteria and data used.
The most lack is about the historical period used, 1961-2000, without explanation because it is not 1971-2010 or 1981-2010 and without explanation because data since 2000 to 2018 are not used. In addition, the prediction of period 2021-2060 has been did without calibration. The period 2001-2018 could be used for calibration model and successively use calibrated model to define future drought events. Without calibration of the model, it is not possible to say if results are correct and adequate or not.
The results are not supported by analyses.
It is not possible to accept in this form.

Reviewer 3 Report
We would like to suggest a rejection for this paper because it is lack of novelty.
1. L97-98: you mentioned that you will project the precipitation and temperature under RCP4.5 and RCP8.5. Then, what is the significance of using both? What do the different projections represent under the two scenarios?
2. L114-115: what do mean about the ‘nine-model ensemble (9ME)’? is it the arithmetic average of the nine CMIP5 models? If so, in line 175, you said ‘we used a Taylor diagram to evaluate the performance of the 9ME (Figure 2)’, but in Figure 2, there are only nine CMIP5 models. In addition, if you want to use the 9ME rather than single SMIP5 model, you need to demonstrate the superiority of the 9ME.
3. L151: for the equation (9), it’s unclear. I found the original equation in the Andeadis et al. (2005) as follows
Andreadis, K. M., Clark, E. A., Wood, A. W., Hamlet, A. F., & Lettenmaier, D. P. (2005). Twentieth-century drought in the conterminous United States. Journal of Hydrometeorology, 6(6), 985-1001.
Round 2
Reviewer 2 Report
Dear author I respect your significant review of the manuscript. Now it is clearer and bibliography is appropriate. Only it is necessary to explain better the caption of figure 3:
“The distribution of bias between observed and uncorrected 9ME (a and b) and bias-corrected 9ME (c and d) monthly mean precipitation (mm/month) and temperature (°C) in China.”
It is not clear what are a and b and what are c and d. Which figure is precipitation uncorrected? Which figure is precipitation corrected? The same for temperature.
Dear Editor,
The paper is substantially improved as description and as bibliography.
Only a last explanation on figure 3, because the caption it is not clear:
“The distribution of bias between observed and uncorrected 9ME (a and b) and bias-corrected 9ME (c and d) monthly mean precipitation (mm/month) and temperature (°C) in China.”
It is not clear what are a and b and what are c and d. Which figure is precipitation uncorrected? Which figure is precipitation corrected? The same for temperature.
After this adjustment it is possible to published the paper.
